# Alteration of the Exhaled Volatile Organic Compound Pattern in Colorectal Cancer Patients after Intentional Curative Surgery—A Prospective Pilot Study

**DOI:** 10.3390/cancers15194785

**Published:** 2023-09-29

**Authors:** Julia Hanevelt, Ivonne J. H. Schoenaker, Richard M. Brohet, Ruud W. M. Schrauwen, Frederique J. N. Baas, Pieter J. Tanis, Henderik L. van Westreenen, Wouter H. de Vos tot Nederveen Cappel

**Affiliations:** 1Department of Gastroenterology and Hepatology, Isala, Dokter van Heesweg 2, 8025 AB Zwolle, The Netherlands; 2Isala Oncology Center, Isala, Dokter van Heesweg 2, 8025 AB Zwolle, The Netherlands; 3Department of Epidemiology and Statistics, Isala, Dokter van Heesweg 2, 8025 AB Zwolle, The Netherlands; 4Department of Gastroenterology and Hepatology, Bernhoven, Nistelrodeseweg 10, 5406 PT Uden, The Netherlands; 5Department of Surgery, University Medical Center Amsterdam, Meibergdreef 9, 1105 AZ Amsterdam, The Netherlands; 6Department of Surgical Oncology and Gastrointestinal Surgery, Erasmus MC, Dr. Molewaterplein 40, 3015 GD Rotterdam, The Netherlands; 7Department of Surgery, Isala, Dokter van Heesweg 2, 8025 AB Zwolle, The Netherlands

**Keywords:** volatile organic compounds, VOCs, electronic nose, Aeonose^TM^, colorectal cancer

## Abstract

**Simple Summary:**

The current diagnostic modalities used during colorectal cancer follow-up appointments are known to have a limited sensitivity to detect recurrent disease. An electronic nose to detect volatile organic compounds (VOCs) in breath is shown to be a promising novel tool in the primary detection of lung cancer and colorectal cancer. To explore the potential role of an electronic nose in follow-up appointments for colorectal cancer patients, we evaluated the effects of surgery with curative intent on the exhaled VOC pattern. This pilot study showed that the VOC pattern changed shortly after surgery, paving the way for use in further clinical trials to address its added value during CRC follow-up appointments, and its ability to detect recurrent disease.

**Abstract:**

As current follow-up modalities for colorectal carcinoma (CRC) have restricted sensitivity, novel diagnostic tools are needed. The presence of CRC changes the endogenous metabolism, resulting in the release of a specific volatile organic compounds (VOC) pattern that can be detected with an electronic nose or Aeonose^TM^. To evaluate the use of an electronic nose in the follow-up of CRC, we studied the effect of curative surgery on the VOC pattern recognition using Aeonose^TM^. A prospective cohort study was performed, in which 47 patients diagnosed with CRC were included, all of whom underwent curative surgical resection. Breath testing was performed before and after surgery using the Aeonose^TM^. A machine learning model was developed by discerning between the 94 pre-and postoperative breath samples. The training model differentiated between the pre-and postoperative CRC breath samples with a sensitivity and specificity of 0.78 (95%CI 0.61–0.90) and 0.73 (95%CI 0.56–0.86), respectively, with an accuracy of 0.76 (95%CI 0.66–0.85), and an area under the curve of 0.79 (95%CI 0.68–0.89). The internal validation of the test set resulted in an accuracy of 0.75 (95%CI 0.51–0.91) and AUC of 0.82 (95%CI 0.61–1). In conclusion, our results suggest that the VOC pattern of CRC patients is altered by curative surgery in a short period, indicating that the exhaled VOCs might be closely related to the presence of CRC. However, to use Aeonose^TM^ as a potential diagnostic tool in the clinical follow-up of CRC patients, the performance of the models needs to be improved through further large-scale prospective research.

## 1. Introduction

Approximately 30% of patients with stage I–III colorectal cancer, and up to 65% of patients with stage IV colorectal cancer (CRC), develop recurrent disease after intentional curative treatment [1]. The early detection of recurrent disease increases the chance of intended curative treatment and survival [2,3]. To date, following-up with patients who have CRC is recommended for up to 5 years after curative treatment, with a more frequent regimen of examinations in the first 2–3 years [4]. Follow-up strategies vary widely, however, most guidelines recommend routine carcinoembryonic antigen (CEA) measurements combined with liver and chest imaging [4,5]. Unfortunately, the sensitivity of these modalities to detect recurrent or metastatic disease is limited. For example, the sensitivity of CEA measurements when detecting local recurrences is 60%, and the sensitivity of computed tomography (CT) imaging or ultrasounds for the detection of metastatic liver disease is 68% and 57%, respectively [6]. As these current methods which detect local or metastatic recurrent disease lack accuracy, novel diagnostic tools are needed.

The presence of CRC changes the overall endogenous metabolism, resulting in the release of a specific composition of volatile organic compounds (VOCs) in the exhaled air [7,8,9]. There are various analytical platforms to analyze these exhaled VOCs, which can be roughly divided into two techniques. Mass Spectrometry (MS) performs a selective quantification of different individual VOCs [10]. An electronic nose (eNose) identifies, through pattern recognition techniques, an entire VOC profile instead of measuring individual VOCs [11].

Even though it is still in its infancy, breath analysis of volatile organic compounds (VOCs) could potentially be a minimally invasive and fast diagnostic tool to detect the first presentations of CRC, as well as recurrences during follow-ups [12,13,14,15,16]. Van Keulen et al. showed that an eNose “the Aeonose^TM^”, which is also used in the present study, could discriminate between patients with CRC and advanced adenomas from healthy controls with an accuracy of 0.84 and 0.73, respectively [12]. The potential use of the Aeonose^TM^ in the follow-up of CRC was first demonstrated by Steenhuis et al. In this pilot study, the Aeonose^TM^ was able to distinguish between patients with recurrent and/or metastatic disease from patients without recurrent and/or metastatic disease, with a sensitivity and specificity of 0.88 (CI 0.69–0.97) and 0.75 (CI 0.57–0.87), respectively; the Aeonose^TM^ exhibited an overall accuracy of 0.81 [15].

Multiple studies indicate that a breath-derived analysis of VOCs might be helpful to detect first presentations of CRC and potentially recurrent/metastatic disease [9,12,13,14,15,16,17]. It is, however, unclear as to whether and when the VOC pattern is altered by curative intended surgery. This study aimed to evaluate the effects of curative surgery on exhaled VOCs in CRC patients through breath-derived VOC pattern recognition using the Aeonose^TM^.

## 2. Materials and Methods

### 2.1. Study Design and Patient Selection

A prospective observational cohort study with a pretest–posttest design was performed in which the VOC patterns of CRC patients, before and after curative surgery, were compared. All patients older than 18 years, diagnosed with CRC (AJCC stadium I–III), and eligible for curative surgical resection, were included. Exclusion criteria included patients with synchronous metastases, who were undergoing neoadjuvant treatment, or patients who had suffered any other type of malignancy (except for basal-cell carcinoma) in the preceding 5 years. Patients that were not competent in the Dutch language or estimated to be physically incapable of performing the breath analysis were also excluded from the study.

Patients were included in two hospitals, as follows: Isala (Zwolle) and Bernhoven (Uden), The Netherlands. The primary outcome measure was the diagnostic accuracy of the Aeonose^TM^ to discriminate between pre- and postoperative breath samples. Breath tests were performed at the time of diagnosis and subsequently during the first post-operative outpatient visit at the department of surgery. Patients were defined as curatively treated when the tumor was resected with histological tumor-free resection margins.

### 2.2. Study Procedures

Before breath samples were taken, exogenous factors that might influence the VOC composition, like smoking, medication, alcohol, or fasting time, as well as endogenous patient characteristics like the patient’s Body Mass Index (BMI) or specific comorbidities, were collected [18]. Patients breathed into the device for 5 min through a disposable mouthpiece fitted with carbon filters that prevent contamination of the inhaled air with environmental VOCs. All patients wore a nose clip during the 5 min of breathing, and they were instructed to close their lips firmly around the disposable mouthpiece to avoid pollution with unfiltered air. In order to avoid confounding factors related to the device, the pre-and postoperative breath tests were performed with the same device. To standardize the execution of the breath test, all healthcare practitioners executing the breath test were instructed during a short instruction class.

### 2.3. Aeonose^TM^ Technology and Model Development with Machine Learning

Breath tests were carried out with three CE-certified Aeonose^TM^ devices from The eNose Company (The eNose Company, Zutphen, The Netherlands). Aeonose^TM^ technology has successfully been used and described for the diagnosis of lung cancer by Kort et al. [11,19,20]. The Aeonose^TM^ contains three micro hotplate metal-oxide sensors that behave like semiconductors. These sensors contain various types of metal and catalyzing agents. The present VOCs in the exhaled breath provoke a redox reaction on the surface of the sensors that subsequently changes the measured conductivity. The redox reactions are dependent on the present VOCs, types of sensors, reaction dynamics, and temperature. The Aeonose^TM^ uses thermal cycling, in which the temperature varies between 260 and 320 °C [21], thus allowing the generation of specific VOC signals. Recording the passing of this thermal cycle with each specific sensor obtains a specific and unique pattern that resembles the measured gas composition. One single breath test generates a data matrix of 64 × 36 conductivity values per sensor. After pre-processing, the data are compressed using singular value decomposition (SVD) to avoid overfitting [11]. Compression of the data then generates one vector with a length of 17. This single vector is used as input to train the different machine learning algorithms and to classify them as accurately as possible, which are then analyzed using the proprietary software program ‘Aethena’ version 2.64 [11].

A machine learning model was developed by differentiating between the total pre-and postoperative breath samples (*n* = 94). The train-test split method was used for model building, with a train-test split ratio of 80–20%. The model was developed using the training set, and its performance was evaluated using the test set. Due to the small sample size, training the model may encounter high variance and it might be more prone to overfitting. Therefore, ‘10-fold cross-validation’ was used to train our machine-learning model. After bootstrapping the performance of each model, they were evaluated and the optimal model was chosen, which was a model based upon an eXtreme Gradient Boosting (XGBoost) algorithm. (Figure 1) To obtain the optimal discrimination performance, the threshold was set to 0.20, meaning that all breath tests with a predictive value of >0.20 were classified as positive for CRC, and those with a predictive value of <0.20 were classified as negative for CRC.

### 2.4. Statistical Analysis

Although it is not possible to calculate the exact sample size in machine learning studies, previous data implied that at least 25 samples are required to develop a solid model in the training phase [12,22,23,24,25,26]. Performance metrics included sensitivity, specificity, accuracy, positive predictive value (PPV), negative predictive value (NPV), and the Area Under the Curve (AUC) of the ‘Receiver Operating Characteristics curve (ROC-curve)’. In addition, the test results were compared before and after surgery by analyzing them in pairs. In this paired analysis, we examined whether the results were similar (concordant) or different (discordant) before and after the surgery.

Demographic data and baseline characteristics were summarized using means and standard deviations for normally distributed continuous data, or median and interquartile ranges for non-normal distributions. Patient and test characteristics between the correct and incorrectly predicted breath samples were compared using the Student’s *t*-test for normally distributed continuous data, the Mann–Whitney U test for non-normal distributed continuous data, and the chi-squared test for categorical data. Normality was verified using the Shapiro–Wilk test. A (two-sided) *p*-value < 0.05 was considered significant. Data analysis of the baseline characteristics was performed using the Statistical Program for the Social Sciences (SPSS) version 28.0 (IBM. Corp, Armonk, NY, USA).

### 2.5. Ethics

The Medical Ethics Review Committee (METC) of Isala, Zwolle, has declared that the above study protocol is not subject to the Medical Research Involving Human Subjects Act (WMO) (Isala METC 210201/CHANGE study). Informed consent was obtained from all participants before sampling.

## 3. Results

### 3.1. Study Population and Baseline Characteristics

In total, 78 patients were eligible for the study, and informed consent was obtained. Ten breath tests failed due to technical problems, mainly connection errors while transferring the data from the iPad to the server. Ten patients were excluded from the study because of a failed breath test due to patients’ discomfort. The most frequently mentioned symptoms, in this case, were excessive coughing and dyspnea. In addition, four patients did not meet the inclusion criteria. One of them appeared to have no malignancy after the pathological examination. Two patients were intraoperatively diagnosed with peritoneal metastases, and another patient received neo-adjuvant chemotherapy because of locally advanced colon cancer. Six patients did not perform a second breath test following surgery due to different reasons, and they were therefore classified as lost during the follow-up appointment. Finally, one patient was diagnosed with urothelial cancer during the study, and was therefore excluded from further analysis (Figure 2).

This resulted in a total of 47 patients who were enlisted to provide samples to develop the model, which occurred by differentiating between the total pre-and postoperative breath samples (*n* = 94). The model was developed using a training data set of 74 samples, and it was internally validated using a test set of 20 samples. Patients who were eligible between March 2021 and March 2022 were included. The baseline patient characteristics of the patient population are shown in Table 1. The mean age was 65 years, and most patients were male (63.8%). The tumors were just as often located on the right side of the colon, as on the left side, and a minority involved rectal cancer (6.4%). Postoperative breath samples were obtained 18 days (median) after intentional curative surgery.

### 3.2. Model Performance

The train and test set scatterplot, in which each individual dot represents the predictive value of all individual pre-operative (green) and post-operative breath tests (red), is shown in Figure 3. As the threshold was set to 0.20, all breath tests with a predictive value of >0.20 were classified by the model as “positive for CRC”. The final XGBoost model was able to discriminate between pre- and postoperative CRC breath samples with a sensitivity and specificity of 0.78 (95% CI 0.62–0.90) and 0.73 (95% CI 0.56–0.86), respectively, and an overall accuracy of 0.76 (95% CI 0.64–0.85). The AUC of the ROC curve after interpolation was 0.79 (CI 0.68–0.89) (Figure 4A). The model performance, using the test set, showed a sensitivity and specificity of 0.90 (95% CI 0.55–1.0) and 0.60 (95% CI 0.26–0.88), respectively, an accuracy of 0.75 (CI 0.51–0.91), and the AUC after interpolation was 0.82 (CI 0.61–1.0) (Figure 4B). For all performance metrics, see Appendix A.

### 3.3. Pre- and Post-Operative Comparisons

Pre- and post-operative paired comparisons are depicted in Table 2. In total, 26 patients (55%) showed concordant test results after pre- and post-operative diagnostic testing, indicating that before and after surgery, both tests predicted correctly. The 45% discordant pairs included 2 patients whose pre-operative and postoperative conditions were incorrectly predicted by the model (4%), 7 patients whose preoperative conditions were incorrectly classified as negative for CRC (15%), and 12 patients whose post-operative conditions were still classified by the model as positive for CRC (26%), despite intentional curative surgery (Appendix A). To explain the discordant test results, patient characteristics and tumor history were verified in more detail (see Table 2).

In total, nine out of forty-seven pre-operative samples were predicted incorrectly by the model, and therefore, they were not classified as positive for CRC. In addition to the finding that significantly more patients used a proton pump inhibitor (PPI) in the incorrectly predicted group, we found no other explanation for incorrect pre-operative prediction, such as limited depth of tumor invasion (T-stage).

Fourteen out of forty-seven postoperative samples were incorrectly predicted as preoperative, and therefore, these results were false positives, resulting in a relatively low specificity. The median time between surgery and the second breath test was shorter in the incorrectly predicted group, however, this difference was not found to be significant (Table 2). We found no relation between smoking status, use of alcohol in the past 24 h, or fasting status. In the correctly and incorrectly predicted group, 42.4% and 50% of the patients had a positive lymph node status during the postoperative pathological examination, respectively. Although nodal involvement was more frequently present in the incorrectly predicted group, this difference did not reach statistical significance. Moreover, the rate of postoperative complications was not significantly higher in the false positive group, as compared with the group comprising correctly predicted samples.

## 4. Discussion

In this pilot study, the use of the Aeonose^TM^ has demonstrated that the VOC pattern of patients with CRC was altered within 3 weeks after curative surgery, suggesting that the exhaled volatile organic compounds are related to the presence of CRC. However, as the performance of the model was not optimal, further research is needed to evaluate the added value of the Aeonose^TM^ during the follow-up appointments of CRC patients.

To our knowledge, this is the first study in which pattern recognition is used to evaluate the effect of curative surgery on exhaled VOCs in CRC patients. Similarly, previous studies regarding CRC patients and other malignancies showed promising results, although these were all carried out using mass spectrometry [27,28,29,30]. Markar et al. showed that by using selected ion flow tube mass spectrometry, propanal levels are significantly elevated in breath samples of CRC patients, and they decrease after surgery. Subsequently, hepatic or peritoneal recurrence during follow-up appointments provoked the upregulation of propanal levels, indicating that propanal may have the potential to be a single-breath biomarker of CRC [17]. Altomare et al. also used mass spectrometry to identify a dataset of 32 VOCs that differed significantly between pre-operative and postoperative CRC patients during follow-up appointments (mean 24 months), confirming the modification of volatile organic compounds in breath after curative surgery [30]. Although their findings support the hypothesis that removal of cancer modifies the metabolism and the exhaled VOC pattern, they found that the “breath print” of disease-free patients did not correspond with healthy controls [30]. Furthermore, the large amounts of VOCs presented play a role in carcinogenesis, indicating that the metabolic changes in CRC patients are complex, and therefore, identifying single VOCs with mass spectrometry may be inferior to VOC pattern recognition using an eNose.

The timeframe for our endogenous metabolism to respond to cancer removal is still unrevealed, and therefore, the optimal timing for postoperative breath sampling was impossible to identify in advance. To minimalize the burden on our patients, we combined the postoperative breath test with the regular outpatient clinic visit at the surgery department. Although not significantly, the median time between surgery and the postoperative breath test was shorter in the incorrectly predicted postoperative breath samples, indicating that these breath tests may have been performed too soon to detect changes. Poli et al. compared VOCs from non-small cell lung cancer (NSCLC) patients before surgery (T0), and one month (T1) and 3 years (T2) after curative surgery using mass spectrometry [27]. VOC levels before and one month after surgery were similar, whereas the results 3 years after surgery, for some VOCs, significantly changed. In contrast to these results, Broza et al. revealed that the concentrations of five VOCs, measured using mass spectrometry, significantly decreased within 3 weeks after curative surgery in a small cohort of patients with lung cancer [28]. This may indicate that changes in VOC concentrations could occur shortly after surgery, and therefore, may be closely related to the removal of the cancerous tissue.

This study was limited by its small sample size and study design, and it lacked control groups. The discriminative performance of our model, and its relation to the presence of CRC, should therefore be interpreted with caution. The changes between the pre-and postoperative breath samples might not be solely due to the removal of the cancerous tissue, as other factors are also likely to play a role, such as physiological post-operative inflammatory reactions and perioperative medication. All patients received perioperative antibiotic prophylaxis, comprising a single dose of 500 mg metronidazole and 2000 mg cefazolin, which could have modified the intestinal microbiome, and therefore, the exhaled VOC patterns. Moreover, it is unknown to what extent a segmental colectomy influences the exhaled VOC pattern. Compared with other studies that were carried out with the Aeonose^TM^, we gathered considerably more failed breath tests due to patients’ discomfort [12,15]. This could be explained by the fact that the preoperative breath tests were carried out while patients visited the multidisciplinary oncologic outpatient clinic, where they also received the findings of the dissemination investigations and possible treatment options. It is quite possible that these outcomes brought up a lot of emotions, tension, and stress that might have influenced the performance of the breath test. It is notable that patients who successfully performed a breath test reported almost no discomfort (median discomfort of 1.0 on a scale from 0 to 10). Of the patients that prematurely interrupted the breath test, almost all of them quit within the first minute. This might indicate that it is not a matter of stamina or condition, and therefore, reducing the duration of the breath test will not influence the success rate.

The use of PPI affects the gut microbiome [31,32]. The changes in microbial activity influence the composition of VOCs [18]. In the pre-operative group, significantly more patients used PPI in the incorrectly predicted samples, whereas in the post-operative group, PPI use was higher in the correctly predicted group; however, this difference was not significant. Therefore, a conclusive association between PPI use and incorrect predictions made by the model could not be found in our analysis. As we found no explanation for the discordant test results concerning other patient or tumor characteristics, the calibration of the eNose might be substandard in terms of its ability to discriminate between VOC patterns. Therefore, improvements are needed for further large-scale research with the Aeonose^TM^.

## 5. Conclusions

The results of this pilot study suggest that the exhaled VOC pattern of CRC patients is altered in a short period of time after curative surgery. This finding might indicate that the exhaled VOC pattern is related to the presence of colorectal cancer, justifying further clinical trials, addressing its added value during CRC follow-up appointments to detect recurrent disease. Although the detection of VOC patterns in exhaled breath using an electronic nose is still a novelty, it is a promising non-invasive low-cost technique. Before VOC pattern recognition can be incorporated in the follow-up of colorectal cancer in the future, further large-scale research is required to validate its performance.

## Figures and Tables

**Figure 1 cancers-15-04785-f001:**
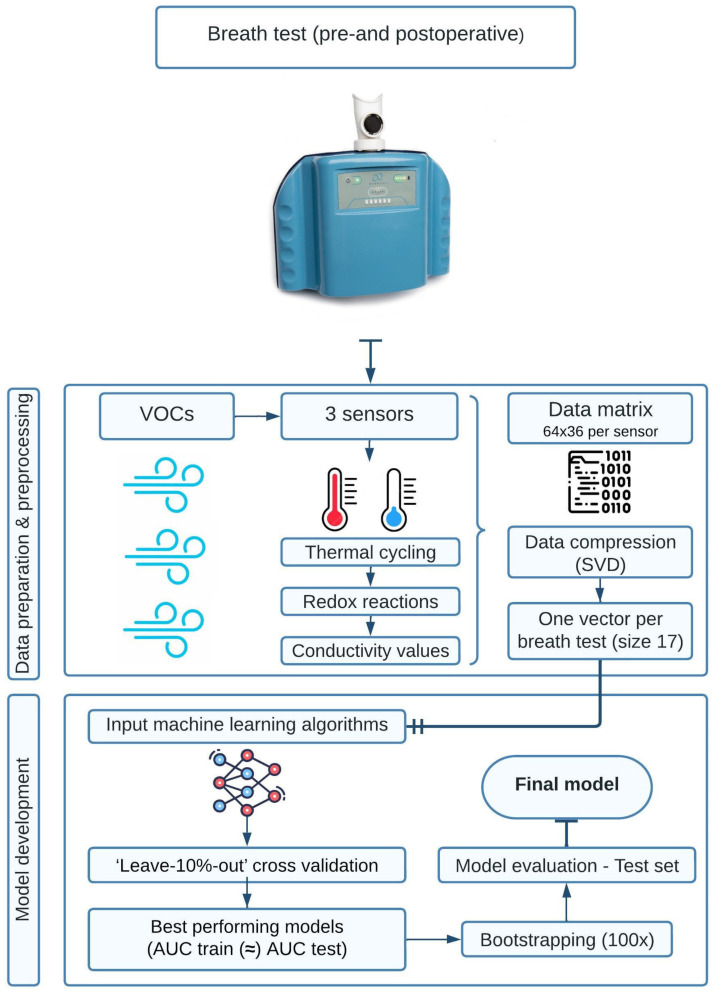
Electric nose (Aeonose^TM^) technology and model development. Abbreviations: VOCs—volatile organic compounds, SVD—singular value decomposition, AUC—area under the curve.

**Figure 2 cancers-15-04785-f002:**
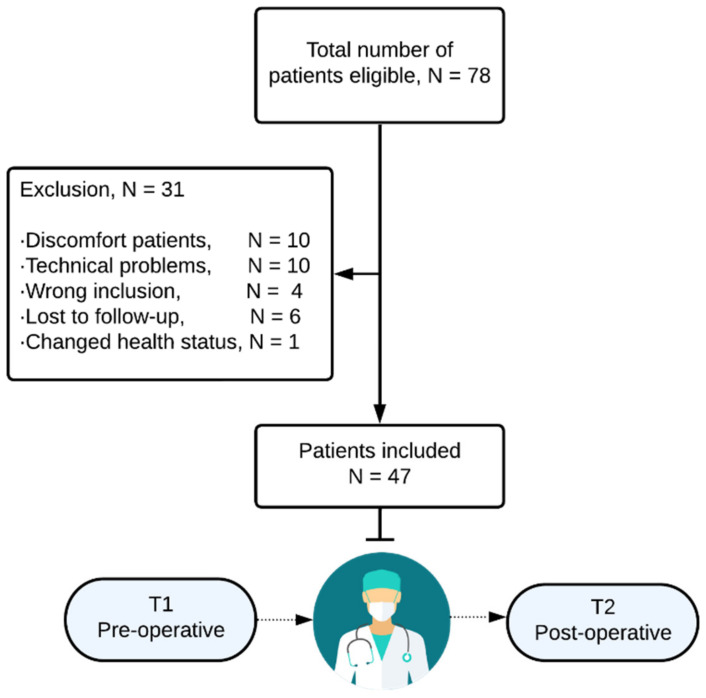
Patient enrollment.

**Figure 3 cancers-15-04785-f003:**
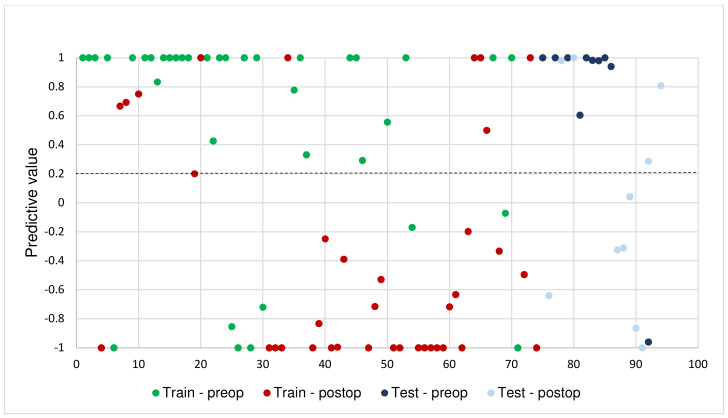
Scatterplot of the individual predictive values of pre-operative (green/dark blue) breath tests and post-operative (red/light blue) breath tests. The threshold was set on 0.20 (dotted line). All breath tests with a predictive value of >0.20 were classified as positive for CRC.

**Figure 4 cancers-15-04785-f004:**
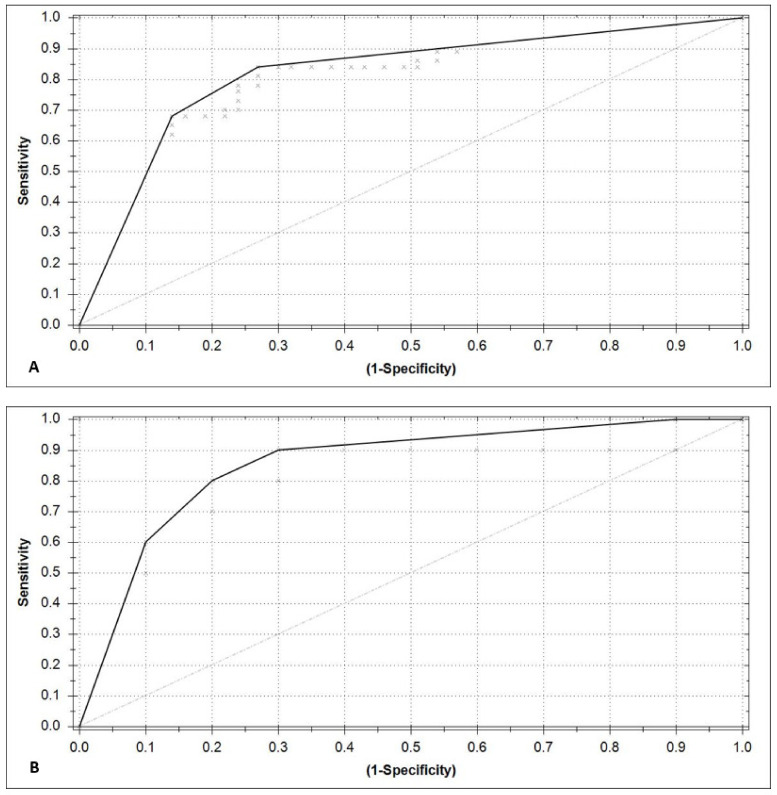
Receiver operating characteristic (ROC)-curve training (**A**) and test set (**B**) after interpolation, representing the diagnostic value of the Aeonose^TM^ in terms of its ability to discriminate between pre-and postoperative breath samples.

**Table 1 cancers-15-04785-t001:** Baseline characteristics.

	Training *n* = 37	Validation *n* = 10	Total *n* = 47
**Patient characteristics**			
**Age, mean ± SD**	64 ± 11.2	68 ± 7.4	65 ± 10.5
**Gender, male, *n* (%)**	23 (62.2)	7 (70.0)	30 (63.8)
**BMI, kg/m^2^, mean ± SD**	25.7 ± 4.1	25.7 ± 2.8	25.7 ± 3.8
**ASA, *n* (%)**			
1	7 (18.9)	2 (20)	9 (19.1)
2	25 (67.6)	7 (70)	32 (68.1)
3	5 (13.5)	1 (10)	6 (12.8)
**Comorbidity, *n* (%)**			
Hypertension	11 (29.7)	1 (10)	16 (34.0)
Cardiovascular comorbidity	5 (13.5)	1 (10)	6 (12.8)
Lung comorbidity	6 (16.2)	1 (10)	7 (14.9)
Thyroid disease	1 (2.7)	-	1 (2.1)
Diabetes	3 (8.1)	1 (10)	4 (8.5)
Hypercholesterolemia	3 (8.1)	1 (10)	4 (8.5)
Autoimmune disease			
IBD	1 (2.7)	-	1 (2.1)
Rheumatoid arthritis	2 (2.7)	-	2 (4.3)
Other			
Reflux esophagitis	3 (8.1)	-	3 (6.4)
OSAS	-	1 (10)	1 (2.1)
Gout	-	1 (10)	1 (2.1)
Depression	1 (2.7)	-	1 (2.1)
Epileptic Disorder	1 (2.7)	-	1 (2.1)
** Tumor characteristics, *n* (%) **			
**Primary tumor location**			
Right colon	14 (37.8)	9 (90)	23 (48.9)
Left colon	22 (59.5)	1 (10)	24 (48.9)
**Rectum**	3 (8.1)	-	3 (6.4)
**Synchronous double tumor**	2 (5.4)	-	2 (4.3)
**MMR-status, *n* (%)**			
MSS	23 (62.2)	5 (50)	28 (59.6)
MSI	1 (2.7)	-	1 (2.1)
Undetermined	12 (32.4)	4 (40)	18 (38.3)
**AJCC (8th edition) stage, *n* (%)**			
Stage I (T1-2N0M0)	7 (18.9)	6 (60)	13 (27.7)
Stage II (T3-T4N0M0)	12 (32.4)	1 (10)	13 (27.7)
Stage III (T1-4N1-2M0)	18 (48.6)	3 (30)	21 (44.7)
**Operation, *n* (%)**			
(Extended) Right hemicolectomy	14 (37.8)	8 (80)	22 (46.1)
Left hemicolectomy	5 (13.5)	-	5 (10.6)
Sigmoid colectomy	12 (35.1)	1 (10)	14 (29.8)
Low-anterior resection	4 (8.5)	-	4 (8.5)
Local resection (TEM & CAL-WR)	1 (2.7)	1 (10)	2 (4.3)
Proctocolectomy	1 (2.1)	-	1 (2.1)
**Time between operation and 2nd test** (**days**)**, median [range]**	20 (10–113)	15.5 (11–76)	18 (10–113)

Abbreviations: BMI—body mass index; ASA—American society of Anesthesiologists; IBD—inflammatory bowel disease; OSAS—obstructive sleep apnea syndrome; MMR—mismatch repair; AJCC—American Joint Committee on Cancer; TEM—Transanal Endoscopic Microsurgery; CAL-WR—Colonoscopic-Assisted Laparoscopic Wedge Resection.

**Table 2 cancers-15-04785-t002:** Paired comparison of correctly predicted versus incorrectly predicted breath samples.

	Pre-Operative			Post-Operative		
	Correctly Predicted *n* = 38	Incorrectly Predicted *n* = 9	*p*-Value	Correctly Predicted *n* = 33	Incorrectly Predicted *n* = 14	*p*-Value
**Age, mean ± SD**	64 ± 10.7	69 ± 9.5	0.22	65 ± 11.5	64.5 ± 8.6	0.80
**Gender, male, *n* (%)**	25 (65.8)	5 (55.6)	0.56	21 (63.6)	9 (64.3)	0.97
**BMI, kg/m^2^ ± SD**	25.8 ± 4.0	25.5 ± 2.8	0.85	26.1 ± 3.7	24.3 ± 3.5	0.13
**Current smoker, *n* (%)**	9 (23.7)	1 (11)	0.41	4 (12.1)	1 (7.1)	0.61
**Alcohol use < 24 h, *n* (%)**	22 (57.9)	3 (33.3)	0.18	8 (24.2)	4 (28.6)	0.76
**Use of corticosteroids, *n* (%)**	4 (10.5)	1 (11.1)	-	3 (9.1)	2 (14.3)	-
**Use of antibiotics ^a^, *n* (%)**	2 (5.3)	1 (11.1)	-	3 (9.1)	1 (7.1)	-
**PPI use, *n* (%)**	8 (21.1)	5 (55.5)	0.04	12 (36.4)	2 (14.3)	0.13
**Last meal < 3 h, *n* (%)**	17 (44.7)	5 (55.5)	0.56	23 (69.7)	10 (71.4)	0.91
**Depth of tumor invasion, *n* (%)**						
**pT1-2**	14 (36.8)	3 (33.3)	0.84			
**pT3-4**	24 (63.2)	6 (66.7)				
**Postoperative complications, *n* (%)**				6 (18.8)	4 (26.7)	0.54
**Minor complication (grade I–II ^b^)**				5	3	-
**Major complication (grade ≥ III ^b^)**				1	1	-
**Polyps in place during second test, *n* (%)**				13 (39.4)	3 (21.4)	0.24
**Time between operation and 2nd test (days), median, [range]**				20 (10–113)	17 (11–76)	0.23

Abbreviations: BMI—body mass index; PPI—proton pump inhibitor. ^a^ Use of antibiotics within 3 months prior to breath test. ^b^ According to the Clavin–Dindo Classification.

## Data Availability

Data are available upon request.

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
