# Peer review of "Alteration of the Exhaled Volatile Organic Compound Pattern in Colorectal Cancer Patients after Intentional Curative Surgery—A Prospective Pilot Study"

_cancers, 2023, doi:10.3390/cancers15194785_

Round 1
Reviewer 1 Report
The authors present a study to detect presence of colorectal cancer (CRC) in patients' breath samples before and (3 weeks) after surgery by using an electronic nose (Aeonose, miniaturised metal oxide sensors). The sensor response differs on breath samples obtained before and after operation. This is believed to be due to increase and decrease of levels of certain VOCs, one of them probably being propanal. Thereby electronic nose data on breath samples before and after CRC operation could indicate the success of chirurgical intervention in an non-invasive way using patients' breath samples.
The predictive result is however only revealed after application of multiple methods of data processing, whereby it is not clear to me, what the effect is of each step of data processing, and whether results might be different if data processing steps were omitted. This is of special concern as the results obtained are not very conclusive (55% correct predictions) and the number of patients (47) is quite low. Moreover, the patients also suffer from different conditions, which complicates interpretation of results. With this outcome, it is crucial to question data processing methods.
In a previous study (15) the outcome obviously was more promising that the results in this manuscript.
Regarding data processing, I would like to see the results obtained without singular value decomposition, 10-fold cross-validation and for other values of discrimination than 0.20.
Furthermore, the following points need to be corrected:
- Lines 21 and 23: should read: electronic nose
- Line 131: a train-test ratio of 0.8 - 0.2% seems not to be very meaningful for neural network analysis. Please comment why you have chosen such a low ratio or correct the number
- Line 219: only 55% of patients showed concordant test results after pre- and post-operative diagnostic Aeonose testing, which seems to be quite low or even random for diagnostic application of the method. Please comment and supply arguments why the method should be applicable in a useful way.
- Line 276: I wonder if the outcome of the study would be better if the breath sampling were performed generally 3 weeks after surgery for better comparison of condition. Do you have more data available now to include in the manuscript?
- Line 369: should read: van Keulen, KE
- Line 399: reference incomplete, add page numbers
Author Response
ANWERS TO THE REVIEWER
We would like to thank the editorial board of Cancers for the opportunity to submit our revised manuscript according to the remarks of the reviewers. We appreciate the time and effort that you and the reviewers dedicated to provide feedback on our manuscript and are grateful for the insightful comments and valuable improvements to our paper. In the point-by-point reply below, all the remarks of the reviewers have been addressed. Both reviewers stated that the English language was fine and that there were no issues detected.
Comments - Reviewer 1
The authors present a study to detect presence of colorectal cancer (CRC) in patients' breath samples before and (3 weeks) after surgery by using an electronic nose (Aeonose, miniaturised metal oxide sensors). The sensor response differs on breath samples obtained before and after operation. This is believed to be due to increase and decrease of levels of certain VOCs, one of them probably being propanal. Thereby electronic nose data on breath samples before and after CRC operation could indicate the success of chirurgical intervention in an non-invasive way using patients' breath samples.
- a) The predictive result is however only revealed after application of multiple methods of data processing, whereby it is not clear to me, what the effect is of each step of data processing, and whether results might be different if data processing steps were omitted. This is of special concern as the results obtained are not very conclusive (55% correct predictions) and the number of patients (47) is quite low. Moreover, the patients also suffer from different conditions, which complicates interpretation of results. With this outcome, it is crucial to question data processing methods. In a previous study (15) the outcome obviously was more promising that the results in this manuscript. Regarding data processing, I would like to see the results obtained without singular value decomposition, 10-fold cross-validation and for other values of discrimination than 0.20.
Furthermore, the following points need to be corrected:
- b) Lines 21 and 23: should read: electronic nose
- c) Line 131: a train-test ratio of 0.8 - 0.2% seems not to be very meaningful for neural network analysis. Please comment why you have chosen such a low ratio or correct the number
- d) Line 219: only 55% of patients showed concordant test results after pre- and post-operative diagnostic Aeonose testing, which seems to be quite low or even random for diagnostic application of the method. Please comment and supply arguments why the method should be applicable in a useful way.
- e) Line 276: I wonder if the outcome of the study would be better if the breath sampling were performed generally 3 weeks after surgery for better comparison of condition. Do you have more data available now to include in the manuscript?
- f) Line 369: should read: van Keulen, KE
- g) Line 399: reference incomplete, add page numbers
ANSWER(S):Dear reviewer,
We would like to thank you for your time and critical assessment of our manuscript. We hope to clarify your other comments/questions point-by-point below.
a) Thank you for your question and suggestions. The data processing steps taken are required in order to train a model and validate the test set. Data compression (in this project: singular value decomposition) reduces the number of free parameters to an amount that is lower than the number of patients to prevent overfitting. Applying cross validation is required to make sure the algorithm is being trained on distinguishing between sick and healthy participants, and not on some artefact. So, you can’t skip data compression or cross validation. With respect to other values of discrimination than 0.20: the value of 0.20 is chosen in the scatter plot (fig.3) in such a way that sensitivity and specificity are 78% and 73% respectively. This combination of sensitivity and specificity can be found as a position in the ROC plot. Changing the discrimination value changes the position in the ROC plot and thus sensitivity/specificity. Setting the threshold on 0.20, we balanced having as little false-negative outcomes as possible with as many true-positives as possible. This is therefore the most optimal threshold. In conclusion, modification of the used methods is not possible and would not improve the performance of the model. The model, however, definitely needs improvement, as it is indeed true 55% correct pre-and postoperative predictions is low. The number of patients was too low to detect associations between the comorbidities and test outcomes. Nevertheless, this pilot study provides more insight into the use of an electric nose in the diagnosis and follow-up of CRC, highlighting the pitfalls and the fact that the model’s performance needs improvement before it can be used in daily clinical practice.
b) This is corrected in the revised manuscript.
c) This should be 80% - 20%, we corrected the revised manuscript
d) We agree that 55% concordant test results are quite low, indicating that the model definitively needs improvement. However, this pilot study was performed to observe the changes in the VOC pattern before and after surgery, in order to explore the potential role of an eNose in the follow up of CRC. The diagnostic performance is far from accurate at this moment, emphasizing further prospective large clinical trials that evaluate its value in follow-up, as already stated in discussion lines 356-359.
As detection of CRC through an electric nose is still premature, we believe that experimental pilot studies like our own, do contribute to the evidence that needs to be provided before large scale studies will be carried out. Several small changes were made in the revised manuscript to underline these thoughts and nuance our conclusions. (see tracked changes in the summary, abstract, discussion).
e) We agree that it would have been better to standardize the timing of the postoperative breath test. Unfortunately, it was not possible to obtain all of the test 3 weeks after surgery due to several postoperative complications and logistical issues. We have no additional data available, so further research should clear up the timeframe that it takes our metabolism to ‘’normalize’’ after CRC and its surgery, including control patients. It might be better to compare these patients on the standard follow-up moments after surgery.
f) This is corrected in the revised manuscript.
g) The page numbers of the reference are added.
Reviewer 2 Report
This paper deals with the hot topic of cancer detection/monitoring by the analysis of the exhaled breath. A e-nose is used in this study.
The paper is very interesting and written, however there are several points which need to be addressed.
1. I am not sure you can define the enose outcome as a VOCs pattern since they are not individualized. E-nose sensors are reactive to class of chemical compounds not specific VOCs
2. Why proton pump inhibitors should influence the VOCs pattern
3. While the median interval between operation and postoperative breath sample is 18, some of the breath samples were get immediately after surgery when the postoperative stress may play a relevant role on the metabolism. This point is relevant and must be justified
4. In the discussion the limited value of this study must be highlighted because of the low sensitivity and specificity, the small sample size, the lack of a control group both before and after surgery. In fact a previous study using an e nose was unable to discriminate healthy subjects from patients with colon cancer and polyps (ech Coloproctol. 2016 Jun;20(6):405-409. doi: 10.1007/s10151-016-1457-z. ).
5. Figure 3 must be re-drawn because the colors used do not allow distinction between different outcome
Author Response
ANWERS TO THE REVIEWER
We would like to thank the editorial board of Cancers for the opportunity to submit our revised manuscript according to the remarks of the reviewers. We appreciate the time and effort that you and the reviewers dedicated to provide feedback on our manuscript and are grateful for the insightful comments and valuable improvements to our paper. In the point-by-point reply below, all the remarks of the reviewers have been addressed. Both reviewers stated that the English language was fine and that there were no issues detected.
Comments - Reviewer 2
This paper deals with the hot topic of cancer detection/monitoring by the analysis of the exhaled breath. A e-nose is used in this study.
The paper is very interesting and written, however there are several points which need to be addressed.
- I am not sure you can define the enose outcome as a VOCs pattern since they are not individualized. E-nose sensors are reactive to class of chemical compounds not specific VOCs
- Why proton pump inhibitors should influence the VOCs pattern
- While the median interval between operation and postoperative breath sample is 18, some of the breath samples were get immediately after surgery when the postoperative stress may play a relevant role on the metabolism. This point is relevant and must be justified
- In the discussion the limited value of this study must be highlighted because of the low sensitivity and specificity, the small sample size, the lack of a control group both before and after surgery. In fact a previous study using an e nose was unable to discriminate healthy subjects from patients with colon cancer and polyps (ech Coloproctol. 2016 Jun;20(6):405-409. doi: 10.1007/s10151-016-1457-z. ).
- Figure 3 must be re-drawn because the colors used do not allow distinction between different outcome
ANSWER(S):
Dear reviewer,
We would like to thank you for your time and critical assessment of our manuscript. We hope to clarify your other comments/questions point-by-point below.
- It is correct that e-nose sensors do not detect all kinds of VOCs in exhaled breath. In the case of (three different) metal-oxide sensors as used in the Aeonose, only VOCs that are capable of a redox-reaction can be analyzed. It is still a VOC-pattern, but the pattern is limited to compounds that can be oxidized.
2. Previous studies demonstrated that use of proton pump inhibitors (PPI) affects the gut’s microbiome - Bosch et al. (ref 18), Imhann Gut 2016, Jackson Gut 2016 (new references added). It is well known that microbial activity influences the composition of VOCs, which would therefore be the most straightforward explanation. In the pre-operative group, more patients used PPI in the incorrectly predicted samples, whereas in the post-operative group, PPI-use was higher in the correctly predicted group, although this difference was not significant. (Table 2) An conclusive association between PPI use and wrong prediction by the model, could therefore not be found in our analysis. A section on this topic was added to the discussion (lines 308-313).
3. The minimum amount of days between surgery and the postoperative breath test was 10 days (Table 2). Postoperative medication, inflammation and stress could have definitely influenced the VOC pattern of patients that did the postoperative breath test ‘’early’’. This is stated in the discussion section, lines 289-292.
4. We agree that it is hard to draw a firm conclusion based upon our study results. The sample size is limited, as well as the lack of control groups. This was highlighted in the discussion, lines 287-288.
The study you mention was performed with another enose (electronic nose PEN3, Airsense Analytics GmbH, Schwerin, Germany) and included a small number of patients with CRC (n = 15). In contrast to their results, van Keulen et al. ref 12, retrieved better results in detecting CRC (AUC of 0.84, sensitivity 95% and specificity 64%) in a larger cohort (n = 70). In conclusion, we believe that the use of eNose in the detection and follow-up in CRC is promising, although further large scale prospective research is definitively needed before it can be further implemented, as stated extensively throughout the revised manuscript.
5) Thank you for the comment. The colors of figure 3 were changed.
Round 2
Reviewer 1 Report
All issues were addressed in the revised version of the manuscript.
Author Response
Dear reviewer,
We would like to thank you again for your time and critical assessment of our manuscript. We are glad that all your comments and questions were clarified and addressed appropriately.
Reviewer 2 Report
The paper has been certainly improved after revision. My main doubt is the time interval between surgery and postoperative breath which was increased from a minimum of 0 days to 10. this a critical issue and the Authors should justify this change in the text
Author Response
Dear reviewer,
We would like to thank you again for your time and critical assessment of our manuscript. We are glad that your previous comments and questions were addressed appropriately, which improved the quality of our manuscript.
To clarify, in the original manuscript, we described the time between surgery and the second breath test as median with interquartile range e.g. 18 [0-16]. In this case, the minimum days between surgery and the second breath test was already 10 days, however we presented it as the interquartile range starting with 0. In the revised manuscript, we presented this variable as median with range e.g. 18 [10-113], as this provides more detailed information.
For clarification, the time between surgery and the second breath test did not change during revision. We only changed the way of data presentation (interquartile range -> range).